# Bacterial Nanocellulose from *Komagataeibacter Medellinensis* in Fique Juice for Activated Carbons Production and Its Application for Supercapacitor Electrodes

**DOI:** 10.3390/polym15071760

**Published:** 2023-04-01

**Authors:** Juliana Villarreal-Rueda, Zulamita Zapata-Benabithe, Laia Posada, Estefanía Martínez, Sara Herrera, Stiven López, Ana B. J. Sobrido, Cristina I. Castro

**Affiliations:** 1Semillero de Termofluidos y Conversión de la Energía, Ingeniería Química, Escuela de Ingenierías, Universidad Pontificia Bolivariana, Medellín 050031, Colombia; 2Grupo de Energía y Termodinámica, Ingeniería Química, Escuela de Ingenierías, Universidad Pontificia Bolivariana, Medellín 050031, Colombia; 3Grupo de Investigación sobre Nuevos Materiales, Ingeniería en Nanotecnología, Escuela de Ingenierías, Universidad Pontificia Bolivariana, Medellín 050031, Colombiacristina.castro@upb.edu.co (C.I.C.); 4School of Engineering and Materials Science, Queen Mary University of London, London E1 4NS, UK; a.sobrido@qmul.ac.uk

**Keywords:** activated carbon, chemical activation, bacterial nanocellulose, supercapacitors, energy storage

## Abstract

This paper presents the results obtained from the chemical activation of bacterial nanocellulose (BCN) using fique juice as a culture medium. BNC activation (BNCA) was carried out with H_3_PO_4_ and KOH at activation temperatures between 500 °C to 800 °C. The materials obtained were characterized morphologically, physicochemically, superficially, and electrochemically, using scanning electron microscopy, X-ray photoelectron spectroscopy (XPS), the physisorption of gases N_2_ and CO_2_ at 77 K and 273 K, respectively, cyclic voltammetry, chronopotentiometry, and electrochemical impedance spectroscopy (EIS). The samples activated with H_3_PO_4_ presented specific surface areas (S_BET_) around 780 m^2^ g^−1^, while those activated with KOH values presented specific surface areas between 680 and 893 m^2^ g^−1^. The XPS analysis showed that the P_XPS_ percentage on the surface after H_3_PO_4_ activation was 11 wt%. The energy storage capacitance values ranged between 97.5 F g^−1^ and 220 F g^−1^ by EIS in 1 M H_2_SO_4_. The samples with the best electrochemical performance were activated with KOH at 700 °C and 800 °C, mainly due to the high S_BET_ available and the accessibility of the microporosity. The capacitance of BNCAs was mainly improved by electrostatic effects due to the S_BET_ rather than that of pseudocapacitive ones due to the presence of phosphorus heteroatoms.

## 1. Introduction

The increasing of global population is followed by a marked rise in energy consumption, primarily from fossil fuels, which has a detrimental effect on environment. In order to struggle these issues, there is a current trend toward adopting alternative technologies to produce “green” energy through the promotion of renewable energy sources such as solar and wind. However, implementing these alternatives is not trivial, mainly because of their intermittency. Therefore, it is critical to guarantee their effective large-scale distribution and storage. In other words, if a large part of the energy comes from renewable sources, there must be a suitable way of storage it and its surpluses, paired with a distribution network with the capacity to transport all the energy generated [1].

Energy storage systems (ESS) come in a variety of forms, such as mechanical (hydroelectric power plants), electrical, thermal (solar thermal power plants), and electrochemical (supercapacitors (SCs) and batteries). Essential parameters in electrochemical ESS are energy density (Wh kg^−1^) to determine how much electrical energy can be stored, and power density (W kg^−1^) to resolve the maximum power the device will deliver. From the Ragone diagram [2], it can be concluded that SCs can store more energy than conventional electrolytic capacitors can, although their power density is lower. On the other hand, the power density of SCs is higher than that of batteries, but the former exhibit lower energy density. SCs store energy by adsorbing ions by a double-layer mechanism [3] and are currently used in applications where fast charging and small amounts of energy are required.

Currently, electrochemical storage systems are getting more attention and having relevance due to the interest in replacing fossil fuels as a primary energy source with more environmentally friendly alternatives, which is why the world is in an “energy transition” process. Due to the low cost, outstanding lifetime and low maintenance of SCs, they are one of the most promising energy storage systems in cases where high power density is needed [4]. SCs are formed by two conductive electrodes spaced apart by an insulating dielectric (usually a polymer) impregnated with a conductive electrolyte (usually a polymer). Applying a voltage to the electrodes causes an accumulation of opposite charges on the material’s surface; those charges are kept apart by the dielectric function, producing an electric field that allows SCs to store electrical energy [5]. There are three different forms of electrochemical energy storage for SCs: (a) pseudocapacitors, which store electrochemical energy transferring electrons between the electrode and an electrolyte; (b) double-layer capacitors, which store energy through the phenomenon of electrostatic charge; and (c) hybrid capacitors, which combine the advantages of the previous two to generate higher energy densities and increase their lifetime [6].

In commercial SCs, carbonaceous materials from non-renewable sources such as graphite or coke carbon serve as primary sources for electrode production. Therefore, it is necessary to look for novel renewable alternatives to replace them and satisfy the currently high demand for sustainability. Activated carbons, carbon nanotubes, and carbon nanofibers are some of the most common materials used to prepare SCs electrodes, each of them presenting different properties and characteristics. For instance, activated carbons with high specific surface areas can be produced from several sources, such as banana peel, palm kernel, sawdust, and others [7]. Additionally, other materials such as aerogels, xerogels, carbon nanotubes, and nanofibers are able to achieve the desired physical and chemical properties for electrodes of SCs, such as moderate conductivity, high thermal resistance, a large specific surface area, and a relatively low cost [8].

There is great interest in using and transforming “green” materials for energy applications. Among these materials is cellulose, one of the most abundant renewable biopolymers in nature, with an outstanding production of around 100 billion metric tons per year, and cellulose-derived carbon materials are ideal for SCs electrodes owing to their high carbon content and tridimensional structure [9,10]. Bacterial nanocellulose (BNC) is a renewable, degradable, and safe-to-use material, as it is a non-toxic material with an interconnected three-dimensional morphology and carbonaceous chemistry. BNC offers significant advantages over vegetable cellulose, with the possibility of large-scale production using raw materials from agroindustrial activity in a relatively straightforward process, which results in a low-cost and ecofriendly material. Additionally, BNC exhibits high crystallinity and mechanical properties of industrial interest, such as a high Young’s modulus (100–130 GPa) and the property of being lightweight (density of 1.6 g cm^−3^), which makes it interesting for applications in portable devices [11]. BNC also offers other physical properties that make it attractive for application as an active material in SCs, including a hierarchical fibrillar structure that promotes a high liquid absorption capacity, high chemical stability, and high mechanical strength [12]. Furthermore, the BNC morphology facilitates the production of active materials with increased interaction between the electrolyte and surface, guaranteeing continuity in ion transmission, which translates into high material performance [13].

Moreover, BNC has attractive properties for application in energy storage devices, such as its high carbon content, ease of surface modification and doping, high specific surface area, and high thermal and structural stability [14]. Some authors have reported the obtention of BNC from bacterial strains such as *Acetobacter xylinum* [15,16,17] or food sources such as nata de coco [18], and its chemically (KOH and H_3_PO_4_) and physically (CO_2_) activation to improve the BNC-specific surface area values (S_BET_) between 340 and 1754 m^2^ g^−1^. The active material has been used as electrochemical double-layer electrodes [18] and catalyst support for the catalytic dehydration of ethanol [16]. For example, Lee et al. [18] reported energy storage capacity values ranging from 26 to 42 F g^−1^ using 0.5 M K_2_SO_4_ as an electrolyte in a potential range between −0.2 V and 0.2 V in a three-electrode cell configuration. Usually, Hestrin-Schramm (HS), a nutrient culture medium obtained via oxidative fermentation, is employed to obtain BNC. However, fique juice is an agroindustrial waste with high levels of nitrogen and carbon in the form of glucose, and it is therefore considered an excellent substrate for BNC production [19]. This work presents the results from the activation of BNC obtained by *Komagataeibacter Medellinensis* using fique juice as a culture medium. The activated carbons chemically obtained are evaluated as supercapacitor electrodes in a two-electrode electrochemical cell in an aqueous medium.

## 2. Materials and Methods

### 2.1. Chemical Activation of the Bacterial Nanocellulose

The BNC was produced by the *K. Medellinensis* strain using fique juice, from the northeast region of Antioquia, Colombia, as a culture medium in accordance with the methodology presented by Castro et al. [20], and dried by lyophilization [19]. The chemical activation of BNC (BNCA) was carried out by impregnation with an activating agent (H_3_PO_4_ 85% or KOH, supplied by Merck (Colombia/Bogotá)) for 24 h at a 1:1 dry weight ratio. The impregnated BNC was dried at 100 °C for 24 h and then heat-treated in a horizontal tube oven in a nitrogen environment at a flow rate of 100 mL min^−1^ at different temperatures: 500 and 600 °C for H_3_PO_4_ (BNCA-P), and 600, 700 and 800 °C for KOH (BNCA-K). The heat treatment was carried out at two heating ramps: the first one was carried out at 3 °C min^−1^ from room temperature to 300 °C for 60 min, and the second one was carried out at 5 °C min^−1^ from 300 °C up to the final temperature for 60 min. Finally, the activated BNC was successively washed with distilled water until it reached a constant pH, and then it was dried in an oven at 100 °C for 24 h.

### 2.2. Morphological, Superficial, Porous, and Physicochemical Characterization

The morphological and compositional characterization of the obtained materials was carried out by scanning electron microscopy (SEM-EDX) using Thermo Fisher Scientific (Colombia/Bogotá) model Apreo 2 S LoVac 5 kV Field Emission Electron Microscope (FESEM).

The superficial and porous characterization was determined by using the physical adsorption/desorption technique for gases (N_2_ at 77 K and CO_2_ at 273 K) in ASAP 2020 Plus equipment (Micromeritics, Colombia/Medellín) after degassing the samples for 8 h at 120 °C in high vacuum conditions. The specific surface area (S_BET_) was calculated by applying the BET model [21], micropore volume (V_N2_ and V_CO2_) determined by the Dubinin–Radushkevich model [22], the micropore size (D_micro_) which was calculated by applying the Stoeckli equation [23], the mesopore volume (V_meso_) which was calculated by the difference in the volume of nitrogen adsorbed at P/P_0_ of 0.98 and the V_N2_ (Gurvicht’s rule). The pore distribution was determined by the density functional theory (DFT) model and Barrett–Joyner–Halenda (BJH) model from the N_2_ isotherm at 77 K. The true density was determined using an Accupyc II 1340 helium pycnometer (Micromeritics, Colombia/Medellín).

The surface chemistry was determined by confocal Raman spectroscopy on Horiba Jobin Yvon equipment (Colombia/Bogotá), the Labram HR model with a high resolution, a 633 nm laser and 15 s exposure time, and by X-ray photoelectron spectroscopy (XPS) X-ray photoelectron spectroscopy (NAP-XPS) with a PHOIBOS 150 1D-DLD analyzer, using a monochromatic Al-Kα (1486. 7 eV, 13 kV, 100 W) with a step energy of 90 eV for general spectra and 20 eV for high-resolution spectra. The high-resolution spectra were recorded for the C_1s_, O_1s_, and P_2p_ elements. The C_1s_ peak at 284.6 eV was used as a reference.

### 2.3. Electrochemical Characterization

The active material was mixed with a polymeric binder (60 wt% polytetrafluoroethylene, PTFE, dispersed in water, supplied by Merck (Colombia/Bogotá)) and a conductive material (acetylene black, Químicos JM (Colombia/Medellín)) in an 80/10/10 weight ratio. The mixture was left to dry in an oven at 100 °C for 24 h. Then, the electrodes were assembled with 7 mg of the mixture on 12.2 mm diameter graphite paper. The electrodes were impregnated with a 1 M H_2_SO_4_ solution (supplied by Merck (Colombia/Bogotá)) for 72 h. The electrodes were characterized in a Swagelok two-electrode cell in a symmetric arrangement and in a three-electrode configuration between 0 and 0.75 V. The gravimetric capacitance was calculated from the capacitance (F) by different techniques (electrochemical impedance spectroscopy, cyclic voltammetry, and chronopotentiometry) [24,25] and normalized by the mass (g) of the active material. The electrochemical characterization was carried out in a Multi Autolab/M204 (Metrohm, Colombia/Bogotá).

Electrochemical impedance spectroscopy (EIS) is a technique used to characterize the frequency response of a device. The Nyquist plot was carried out in a frequency range between 1 mHz and 100 kHz with a sinusoidal amplitude of ± 10 V. The *C_EIS_* (F g^−1^) is the maximum gravimetric capacitance, which was determined according to Equation (1) at the minimum frequency, and the interfacial capacitance *IC* (μF cm^−2^) is the gravimetric capacitance normalized by the specific surface area according to Equation (2).

(1)$$C_{EIS}=\frac{-Z\prime \prime }{m\omega {\left|Z\right|}^{2}}$$(2)$$IC=\frac{C_{EIS}}{S_{BET}}$$
where Z″ is the imaginary impedance, $\left|Z\right|$ is the complex impedance, ω is the angular frequency and m is the total active material weight (g).

Additionally, important parameters on supercapacitor behavior, such as the equivalent series resistance (ESR), charging time (τ) and phase angle (φ), could be determined from the EIS.

The cyclic voltammetry (CV) technique applies a potential perturbation (scan rate) in a specific potential range. The cyclic voltammetry curves were carried out between 0 and 0.75 V at different scan rates $\left(V_{b}\right)$: 0.5, 1, 2, 5, 10, 25, and 50 mV s^−1^. The kinetic or diffusion limitation can be detected from CV curves [24]. The *C_CV_* (F g^−1^) is the gravimetric capacitance, and it was determined from the voltammetry curves according to Equation (3).

(3)$$C_{CV}=\frac{2\sum i\left(t\right)}{m∆V}$$
where $i\left(t\right)$ is the change in the current as a function of time, ∆V is the potential window and, m is the total active material weight (g). The capacitance value decreases as the scan rate increases because the number of active sites diminish inside the material and the redox reaction is not complete with a higher potential.

The capacitance retention percentage (*R_CV_*) was determined from the *C_CV_* values at different sweep ratios with respect to the *C_CV_* calculated at 2 mV s^−1^ (Equation (4)).
(4)$$R_{CV}\left(\%\right)=\frac{C_{CV,s{mVs}^{-1}}}{C_{CV,2{mVs}^{-1}}}\times 100$$

The chronopotentiometry (CP) technique applies a constant current density and measures the potential (E) with respect to time, allowing it to determine the device’s stability. The charge and discharge curves were performed between a potential window from 0 to 0.75 V at different current densities: 0.156, 0.313, 0.625, and 0.938 A g^−1^. The gravimetric capacity (*C_CP_*) was calculated from the discharge curve according to Equation (5). Additionally, the best samples underwent cyclability analysis at the 0.33 A g^−1^ current density (5000 charge–discharge cycles).

(5)$$C_{CP}=\frac{4I_{c}∆t}{m∆V}$$
where $I_{c}$ is the current density (A g^−1^), ∆t is the discharge time (s), ∆V is the potential range (V) and m is the total active material weight (g).

## 3. Results and Discussion

The yield of activated carbons after the BNC activation process ranged from 5% to 38%. BNC activated with H_3_PO_4_ are represented by the letter P followed by the heat treatment temperature (T = 500 and 600 °C), PT, and those activated with KOH are represented by the letter K followed by the heat treatment temperature (T = 600, 700 and 800 °C), KT.

### 3.1. Morphological Characterization of BNCAs

Figure 1 shows SEM micrographs of freeze-dried BNC. The BNC showed a long fiber structure with diameters between 70 and 100 nm and a true density of 1.09 ± 0.01 cm^3^ g^−1^. After activation, the fibrous structure disappeared and the fibers collapsed. The EDX spectra are presented in the Appendix A.

SEM micrographs of H_3_PO_4_-activated carbons (P500 and P600) are presented in Figure 2. Primary particles in the form of microspheres constituted some of the surface morphology of P500 (Figure 2a), but in other regions of this sample, these particles merged into a continuous network (Figure 2b). The primary particles in Sample P600 were also microspheres but smaller than those in P500 (Figure 2c). However, in some areas (P area in Figure 2d), there was a cluster of different morphology particles in the network that was identified by EDX as phosphorus particles (Appendix A).

In the case of BNC activated with KOH, sample K600 exhibited an interconnected structure formed by open pores in some sections (Figure 3a) and a compact surface with macropores (Figure 3b). Sample K700 presented a nanostructured network with variously sized accessible pores (Figure 3c,d). Sample K800 showed a heterogeneous morphology (Figure 3e,f); in one section, the carbon network was composed by primary particles in small microspheres, and, in another section, a three-dimensional network was formed by accessible pores of a smaller size than those of K700. The rise in the temperature significantly affected the morphology of the activated carbons, especially at 800 °C where K800 apparently showed a weaker structure, similarly to the findings reported by Boongate and Phisalaphong [15].

### 3.2. Superficial and Porous Characterization of BNCAs

N_2_ adsorption/desorption isotherms at 77 K are shown in Figure 4a,b, and CO_2_ adsorption isotherms at 273 K are shown in Figure 4c. Samples P500 and P600 present type IV isotherms (Figure 4a) with a type H3 hysteresis cycle according to IUPAC, which corresponds to that of materials with heterogeneous micropore distribution and the presence of wider micropores (~4 nm). Samples K600 and K700 are classified as type Ia, corresponding to the properties of essentially microporous materials, while sample K800 can be classified as type Ia-IV due to the H4 hysteresis cycle, suggesting the presence of both micropores and mesopores (Figure 4b) [26].

The pore size distribution for samples P500 and P600 (Figure 4d) presented a similar distribution of around 1.2 nm and a wide distribution of mesopores of between 2–20 nm. Activated samples with KOH (Figure 4e) exhibited a similar micropore size distribution in the microporous region (<2 nm), with a strong peak at 0.5 nm, for K800.

Table 1 presents the structural properties of the activated materials. The S_BET_ values ranged from 718 to 893 m^2^ g^−1^, values slightly higher than those found in the literature for this type of material [16,17,18]. The temperature did not have a significant effect on microporous development when H_3_PO_4_ was selected as the activating agent; however, the volume of mesopores increased when using H_3_PO_4_. Additionally, the activated samples with KOH showed a substantial increase in the S_BET_, particularly the sample treated at 800 °C, which can be explained by the intercalation of K between the walls of the carbonaceous structure which promotes changes in the microstructure (including defects, edges, morphology, and pore size) [27], and the formation of hierarchical pores. The micropore size of activated samples with H_3_PO_4_ was 1.26 nm and for the activated samples with KOH, the micropore size was 1 nm. The micropore size distribution was broad and heterogeneous because V_N2_ > V_CO2_; the N_2_ could fill all the micropores during its adsorption at 77 K, and the CO_2_ could only fill the ultramicropores (<0.7–0.8 nm wide) [28]. Figure 4f shows that the activation with H_3_PO_4_ mainly promoted the development of mesopores. P500 presented an average pore diameter of 2.8 nm, while P600 presented an average pore diameter of 7 nm with a wider distribution.

### 3.3. Physico-Chemical Characterization of BNCAs

Figure 5a shows Raman spectra of BNCAs which were used to evaluate the degree of graphitization of surfaces.

Three prominent peaks were observed; The first signal intensity (I_D_) of the D-peak at ~1335 cm^−1^, corresponding to the vibrations of carbon atoms with *sp^3^* bonding assigned to defects and disorder of the lattice and to the low symmetry carbon structure of graphite. The second signal intensity (I_G_) of the G-peak at ~1590 cm^−1^ (I_G_), which is characteristic of graphite, corresponding to the *sp^2^* stretching vibration of carbon atoms in the graphite phase. The third peak was observed at 2710–2724 cm^−1^, which can be attributed to second-order phonons [29]. Table 2 compiles the main physicochemical properties of BNCAs determined by XPS, SEM-EDX and Raman.

The graphitization degree is determined from the ratio of I_D_/I_G_ peak areas, and it is related to the carbon material structure as a measure of the extent of disorder in a carbon layer [30]. The I_D_/I_G_ ratio is slightly lower for BNCA-K (~1.87) than BNCA-P (~2.0), indicating that graphitic carbon formation is associated with a loss of dissociable functional groups. An increased in the I_D_/I_G_ ratio promotes a loose fibrous form, as shown by the SEM images of BNCA-P (Figure 2) [18], caused, in this case, by H_3_PO_4_ activation. The disorder in carbon surface distortions could be related to the microporous structure or disorganized regions near the edges, possibly as a result of the introduction of phosphorus functional groups to the surface [31]. Additionally, P600 showed a high I_D_ intensity and a low full width at the half maximum (FWHM) of the I_G_ (Figure 5b), which suggests that structural disorder is caused by the presence of oxygen and hydrogen atoms.

Figure 6 shows XPS high-resolution spectra of C_1s_, O_1s_, and P_2p_ for samples P500 and P600, and Figure 7 shows the C_1s_, and O_1s_ spectra of samples K600 (a and b), K700 (c and d), and K800 (e and f). These peaks were deconvolved according to the binding energy, corresponding to a specific functional group presented in Appendix A [31,32,33]. The BNCAs presented O/C ratios between 13% and 30%, where P600 presented the highest percentage, suggesting that at 600 °C, surface oxidation came from functional groups of phosphors [32].

The percentage of nitrogen on the surface was lower than 0.6 wt% (0.16–0.6 wt%), which may be associated with the residual impurities. However, samples treated at the lowest temperatures showed the highest percentage (P500 and K600). The phosphorus percentage on the surface was 3.1 wt% and 11.1 wt% for P500 and P600, respectively, which are higher than other biomasses activated with H_3_PO_4_ [34]. The P500 sample showed, in the P_2p_ region, a higher percentage (38%) at ~133.5 eV, which corresponds to the C-PO_3_ or C_2_-PO_2_ functional groups. However, the P600 sample presented an increase in pyrophosphates functional groups at ~135.2 eV (33.5 wt%), and an increase in P_2_O_5_ at ~136 eV (33.7 wt%), while C-PO_3_ or C_2_-PO_2_ groups at ~133.5 eV decreased (12 wt%). This suggests that activation temperature plays a key role in the formation of functional groups at the surface of the resulting carbons.

### 3.4. Electrochemical Characterization of BNCAs

Figure 8a,d show the Nyquist plots for the BNCAs. A semicircle in the high-frequency region is observed, followed by a gradual change in the slope from 45° to 90°, which represents Warburg diffusive behavior and is attributed to the diffusive effects of chemical species along the electrode [35]. In the low-frequency region (<1 Hz), a vertical line is observed and represents capacitive behavior. The P600, K700 and K800 samples presented the best capacitive behavior, while the P500 and K600 samples presented a distorted semicircle associated with difficulty in electrolyte diffusion inside the electrode in the mid-frequency region [36].

The CVs of the K700 and K800 samples showed a quasirectangular, horizontal and symmetrical shape at low (Figure 8e) and high (Figure 8f) scan speeds, which is similar behavior to that of an ideal electrochemical double-layer capacitor (EDLC). The P500 (Figure 8b,c) and K600 (Figure 8e,f) samples presented CVs with a nonrectangular shape and that were deviated from being horizontal, which was related to the diffusion limitations of electrolyte ions in the micropores. The P600 sample presented ideal behavior at 2 mV s^−1^ (Figure 8b) with an additional shoulder at a lower potential, which was associated with pseudocapacitive effects, most likely due to the presence of phosphorus heteroatoms [37,38] on the surface (11.1 wt%). However, at 25 mV s^−1^ (Figure 8c), ideality was lost due to the diffusive limitations of the electrolyte inside the carbonaceous matrix. Part of the electrode surface was inaccessible, and just its external surfaces interacted.

Table 3 presents the main electrochemical properties of the BNCAs in terms of the CV and EIS. The K700 and K800 samples presented the highest values of gravimetric capacitance (*C_EIS_*), 181 F g^−1^ and 219.5 F g^−1^, respectively, values higher than those reported in the literature for materials produced under similar conditions [18], but slightly lower than those reported for some electrodes prepared with doped materials mainly with nitrogen, phosphorus and sulfur heteroatoms (43–509 F g^−1^).

The equivalent series resistance (ESR) value can be calculated in the high-frequency area of the Nyquist plot. The ESR represents the sum of the contact resistance, charge transfer resistance, and diffusion resistance in electrolyte pores during the electrochemical process. The ESR is calculated from the vertical line extrapolation at medium-low frequencies of the Nyquist diagram and its cut on the abscissa [39]. The K800 sample presented the lowest ERS value (2 Ω), which benefits electrochemical double-layer formation, while P500 showed the highest ERS value (23 Ω) mainly due to electrolyte diffusion limitations. The relaxation time (τ) is how quickly a capacitor can be charged or discharged. The K800, K700 and P600 samples exhibited the lowest relaxation times due to their low ESR in the order K800 < K700 < P600. The phase angle (φ) is obtained from the bode diagram at the lowest frequency. K700 showed the lowest phase angle value near −90° and the lowest I_D_/I_G_ value, properties that suggest better conductivity due to high graphitization, and the behavior of an ideal supercapacitor [40,41].

From the EIS and CV curves (Figure 8), no supercapacitor works entirely as an ideal capacitor. Therefore, the electrical equivalent circuit could be represented by a “Randles circuit” that is conformed for a double layer capacitance (C_dl_) component parallel to the Faradaic charge (R_F_) and impedance Warburg (W) components, and these are in series with an electrolytic resistance (R_s_) [24]. The K800 sample presented the highest *C_EIS_* and *IC* value (24.6 µF cm^−2^). This last value is slightly higher than that reported for a clean graphite surface [42] and indicates deeper electrolyte accessibility to micropores. The electrochemical behavior of the P600 sample suggests that the presence of phosphorus functional groups improves surface conductivity. However, phosphorus superficial functional groups (135 eV and 136 eV) did not improve electrode surface wettability by electrolytes or the formation of EDLC. Additionally, the P500 sample had a P_XPS_ of 3.1 wt% and an S_BET_ that was like that of other samples. Nevertheless, it presented low electrochemical performance, which may have been related to its compact external morphology, making it difficult for the electrolyte access the micropores.

Microporosity seems to have the largest effect on capacitance (Figure 9a). According to the pore distribution (Figure 4e), the BNCA-K samples presented narrow pore distribution below 2 nm, with an average micropore size between 0.9 and 1.27 nm, which is higher than the optimal size for hydrated molecules ${SO}_{4}^{-2}$, which is 0.7 nm [43]. Figure 9b,c show that capacity increases as ESR decreases as a consequence of the presence of oxygen and phosphorus surface functional groups, mainly C—O single bonds (532.3 eV). Oxygen superficial functional groups are important for the introduction of pseudocapacitive effects that influence the increase in *IC*.

Based on the data shown in Table 3, Figure 10a depicts a reduction in capacitance retention as a function of the scan rate increasing. The K800 sample presented the highest *R_CV_* (84%) at 25 mV s^−1^, followed by K700 (75%) and P600 (73%). The low behavior of the P500 and K600 samples is attributed to the presence of functional groups, which hinder electrolyte ion diffusion. Appendix A shows the electrochemical performance of the K700 sample characterized in a three-electrode cell configuration. The cyclic voltammetry curves show a quasirectangular, horizontal, and symmetrical shape, with the presence of redox peaks in −0.3 V and 0.3 V being assigned to quinone/hydroquinone groups. Additionally, the charge and discharge curves were symmetrical, suggesting supercapacitor behavior. The capacitance values were higher than those shown in Table 4, verifying that these materials could be considered excellent for electrodes for supercapacitor devices.

Figure 10b presents high stability after the 5000 charge and discharge cycles at 0.313 A g^−1^ for the K700 sample. The charge and discharge curves of the first and last cycles are presented in Figure 10c, which have a quasitriangular and symmetrical shape that indicates ideal conservation. The capacitance retention after 5000 cycles was 66% compared to the first cycle at 0.313 A g^−1^. The increase in capacitance around cycle 3500 was related to the contribution of pseudocapacitance by surface functional groups [52].

## 4. Conclusions

BNC obtained from *K. Medellinensis* using fique juice as a culture medium allowed the obtention of chemically activated carbons with high specific surface areas. The materials were evaluated by different electrochemical techniques, obtaining the highest gravimetric capacitances between 180 and 220 F g^−1^, corresponding to those of the KOH-activated materials at 700 °C and 800 °C. These materials showed higher capacitance retention, and after 5000 cycles, a retention of 66% was still present. Temperature is a crucial parameter for developing microporosity and thus improving electrochemical double-layer formation. H_3_PO_4_ activation allowed a high content of phosphorus heteroatoms to be incorporated into the surface; however, the functional groups formed did not show any significant contribution to the pseudocapacitive effects that could have increased the overall electrode performance.

## Figures and Tables

**Figure 1 polymers-15-01760-f001:**
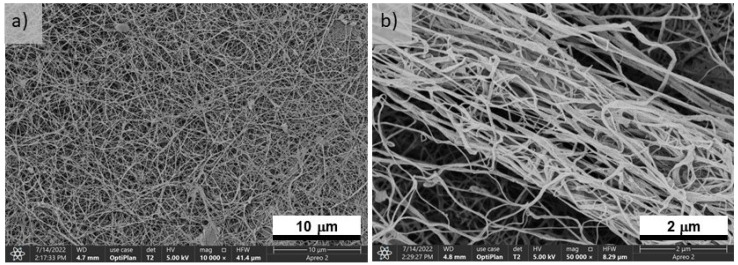
SEM micrographs of freeze-dried BNC at (**a**) ×10,000 and (**b**) ×50,000.

**Figure 2 polymers-15-01760-f002:**
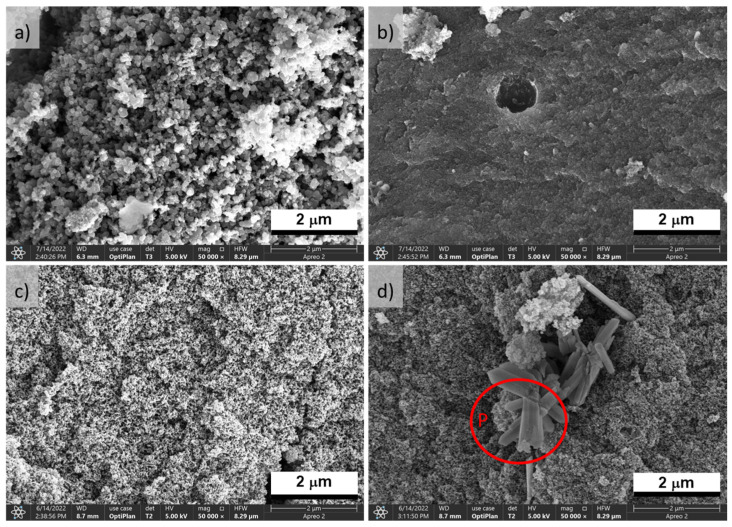
SEM micrographs at ×50,000 of BNCA-Pat 500 °C (**a**,**b**) and at 600 °C (**c**,**d**). P area: clusters of phosphorus particles.

**Figure 3 polymers-15-01760-f003:**
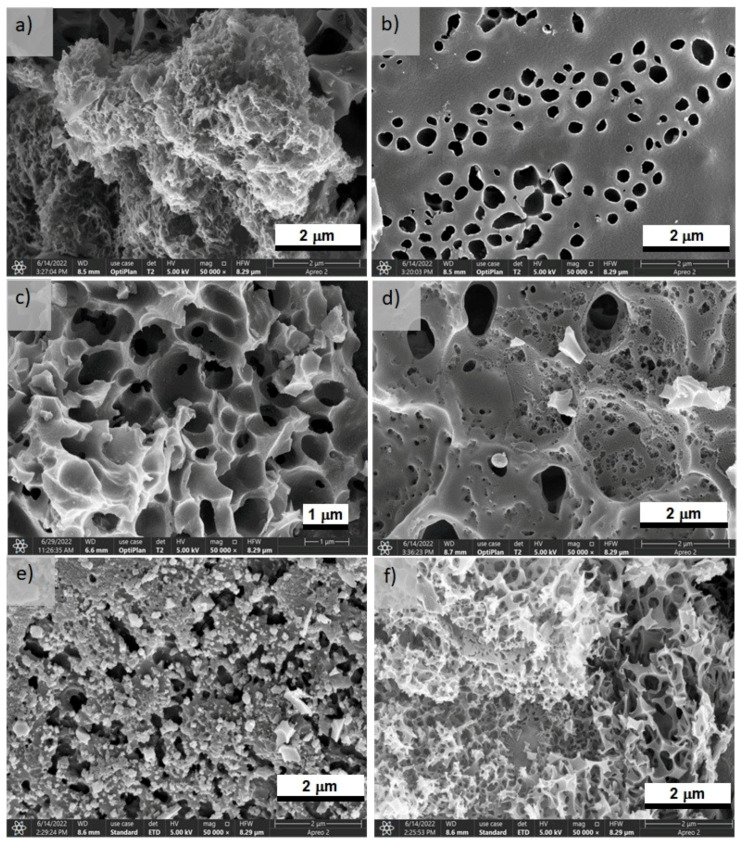
SEM micrographs at ×50,000 of BNCA-K (**a**) at 600 °C (**a**,**b**), at 700 °C (**c**,**d**), and at 800 °C (**e**,**f**).

**Figure 4 polymers-15-01760-f004:**
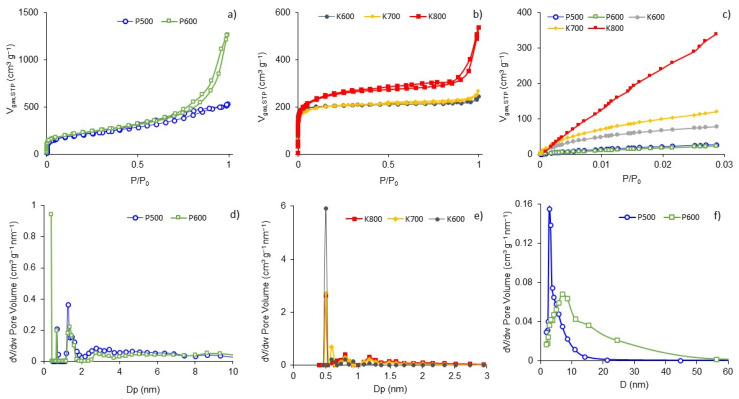
N_2_ isotherms at 77 K for (**a**) BNCA-P and (**b**) BNCA-K; (**c**) CO_2_ isotherms at 273 K, with pore size distribution by DFT for (**d**) BNCA-P and (**e**) BNCA-K, and (**f**) pore size distribution by BJH for BNCA-P.

**Figure 5 polymers-15-01760-f005:**
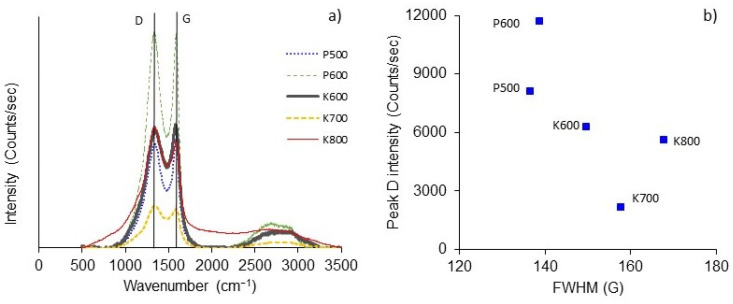
(**a**) Raman spectra of BNCAs, and (**b**) the intensity of I_D_ as a function of the FWHM of I_G_. D-peak: intensity (I_D_); G-peak: intensity (I_G_).

**Figure 6 polymers-15-01760-f006:**
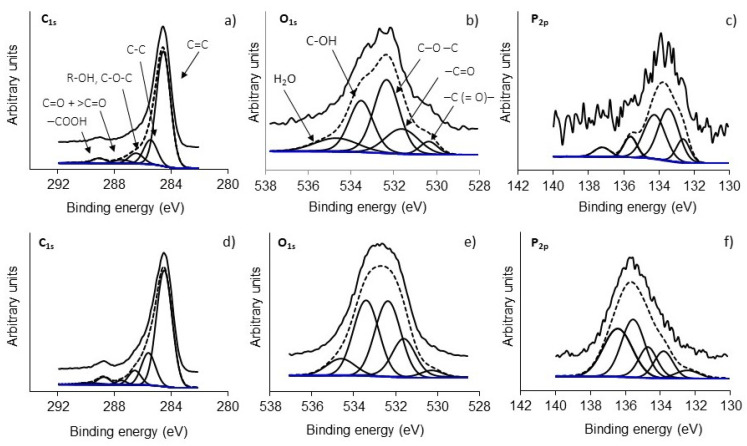
XPS high-resolution spectra of (**a**) C_1s_, (**b**) O_1s_ and (**c**) P_2p_ regions for P500 and (**d**) C_1s_, (**e**) O_1s_, and (**f**) P_2p_ regions for P600.

**Figure 7 polymers-15-01760-f007:**
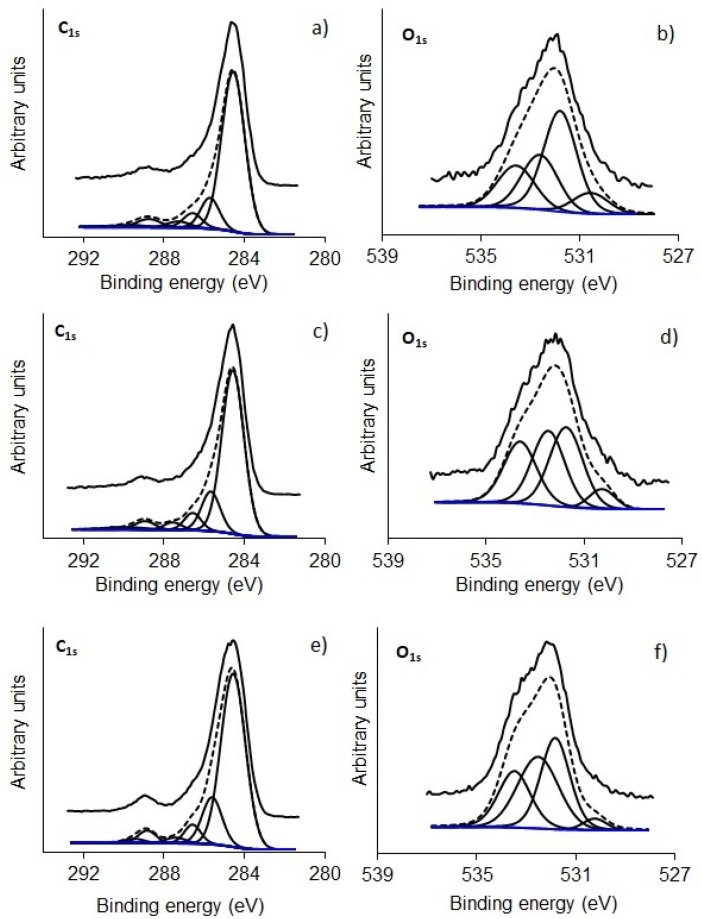
XPS high-resolution spectra of (**a**) C_1s_ and (**b**) O_1s_ regions for K600, (**c**) C_1s_ and (**d**) O_1s_, regions for K700 and (**e**) C_1s_ and (**f**) O_1s_ regions for K800.

**Figure 8 polymers-15-01760-f008:**
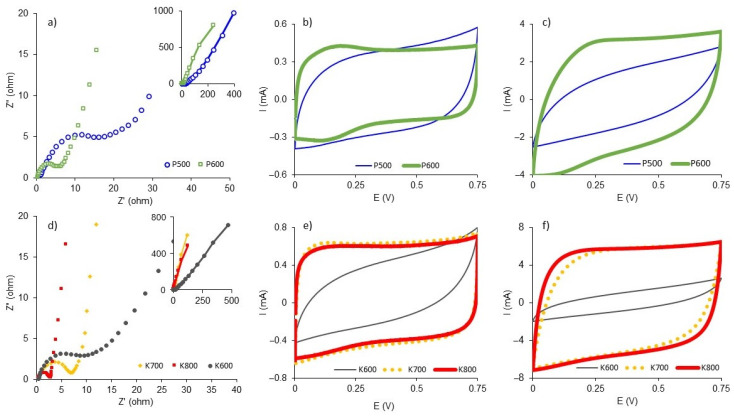
(**a**) Nyquist plot, cyclic voltammetry curves at (**b**) 2 mV s^−1^ and (**c**) 25 mV s^−1^ for samples BNCA-P, and (**d**) Nyquist plot and cyclic voltammetry curves at (**e**) 2 mV s^−1^ and (**f**) 25 mV s^−1^ for samples BNCA-K at 1 M H_2_SO_4_.

**Figure 9 polymers-15-01760-f009:**
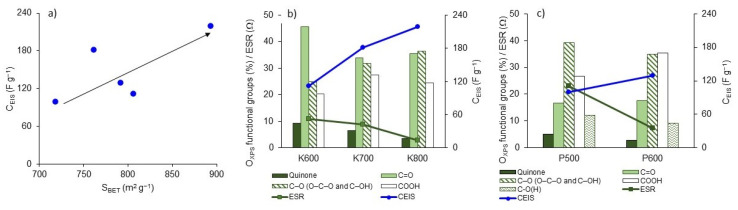
Relationship of gravimetric capacitance (C_EIS_) to (**a**) S_BET_, ESR and oxygen superficial functional groups of (**b**) BNCA-K and (**c**) BNCA-P.

**Figure 10 polymers-15-01760-f010:**
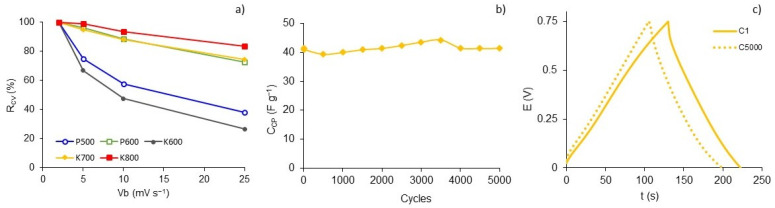
(**a**) Capacitance retention as a function of different scan rates, (**b**) gravimetric capacitance (*C_CP_*) evolution during 5000 charge and discharge cycles at 0.313 A g^−1^ in 1 M H_2_SO_4_ of K700, and (**c**) charge–discharge curves between 0 and 0.75 V of the first and last cycle of K700.

**Table 1 polymers-15-01760-t001:** Structural properties of BNCA.

BNCA	S_BET_ (m^2^ g^−1^)	D_micro_ (nm)	V_N2_ (cm^3^ g^−1^)	V_total,0.985_ (cm^3^ g^−1^)	V_meso_ (cm^3^ g^−1^)	V_CO2_ (cm^3^ g^−1^)
P500	718	1.27	0.234	0.73	0.50	0.084
P600	791	1.26	0.258	1.72	1.46	0.045
K600	806	0.91	0.263	0.33	0.06	0.245
K700	761	1.00	0.236	0.35	0.12	0.177
K800	893	1.22	0.315	0.69	0.38	0.169

**Table 2 polymers-15-01760-t002:** Physicochemical properties of BNCA by XPS, SEM-EDX and Raman.

BNCA	XPS (wt%)					EDX (wt%)				Wavelength (cm^−1^)		I_D_/I_G_
	C_1s_	O_1s_	N_1s_	P_2p_	O/C (%)	C	O	N	P	D	G	
P500	84.7	11.7	0.5	3.1	13.8	31.2	49.6	-	19.2	1336.5	1585.6	2.00
P600	68.2	20.4	0.2	11.1	29.9	40.1	5.1	-	54.7	1333.6	1585.6	2.03
K600	81.9	17.5	0.6	-	21.4	82.4	17.6	-	-	1342.3	1583.7	1.87
K700	84.8	15.0	0.16	-	17.7	70.0	30.0	-	-	1345.2	1590.5	1.86
K800	82.9	16.88	0.21	-	20.4	67.0	33.0	-	-	1337.5	1590.5	1.89

**Table 3 polymers-15-01760-t003:** Electrochemical properties of BNCAs determined by cyclic voltammetry and electrochemical impedance spectroscopy.

BNCA	*C_CV_* (F g^−1^)/*V_b_* (mV s^−1^)				*C_EIS_* (F g^−1^)	ESR (Ω)	−φ (°)	τ (s)	*IC * (μF cm^−2^)
	2	5	10	25					
P500	112	84	64	43	99.8	23.2	68.1	159.2	13.9
P600	107	103	94	78	129.9	7.4	73.3	5.8	16.4
K600	119	79	57	32	111.6	11.0	56.8	159.2	13.8
K700	192	182	169	143	181.6	8.8	78.8	2.3	23.9
K800	180	179	169	151	219.5	2.9	75.7	0.9	24.6

**Table 4 polymers-15-01760-t004:** Gravimetric capacities of electrodes from doped BNCs. 3E: three-electrodes configuration; 2E: two-electrodes configuration.

Reference	Cellulose	Dopant Agent		S_BET_ (m^2^ g^−1^)	Capacitance (F g^−1^)
Wang et al. [44]	Hydrolyzed cotton	*N:*	Urea	123–366	220–275 3E 6 M KOH
Li et al. [45]	BC, Hainan Yide Food Industry Co.	*N, P, and S:*	(NH_4_)H_2_PO_4_, (NH_4_)_2_SO_4_ (NH_4_)H_2_PO_4_/(NH_4_)_2_SO_4_	296–498	80–255 2 M H_2_SO_4_
Liu et al. [46]	BC, Guilin Qi Hong Technology Co.	*N, P, and S:*	Thiourea (CH_4_N_2_S) Rogor (C_5_H_12_NO_3_PS_2_, 40 wt%) Phoxim (C_12_H_15_N_2_O_3_PS, 40 wt%)	261–589	132–509 3E and 2E 0.5 M H_2_SO_4_
Fang et al. [47]	BC, Hainan Yide Food Industry Co.	*O:*	Juglone (5-hy-droxy-1,4-naphthalenedione) Carbon nanotubes	95–102	74.8–461.8 3E 1 M H_2_SO_4_
Wang et al. [48]	BC, Hainan Yeguo Food Company	*N:*	Polypyrrole Carbon nanotubes (20–50 µm)	–	0.15–0.5 F cm^−2^ Gel PVA/H_2_SO_4_
Zhang and Chen [49]	BC, Hainan Yide Food Co.	*N:*	Polydopamine	191–464	26–260 2E and Coin CR2032 1 M H_2_SO_4_
Bai et al. [13]	BC. Yeguo Foods Co.	*O:*	Poly(ethylene-co-vinyl alcohol)	747–2189	420 3E 6 M KOH
Wannasen et al. [50]	BC. *Gluconacetobacter xylinum* en D-glucose	*N, P, Co:*	Cobalt(II) nitrate hexahydrate Phosphoric acid	12–44	43.3 (158.5 mF cm^−2^) 3E 3 M KOH
Xu et al. [51]	BC. Hainan Yide Foods Co., Ltd.	*N:*	1 M Pyrrole	–	459.5 3E 1 M NaCl

## Data Availability

The authors confirm the data transparency included in the manuscript. The data supporting the findings of this study are available within the manuscript and its Appendix A.

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
