# Peer review of "Bacterial Nanocellulose from *Komagataeibacter Medellinensis* in Fique Juice for Activated Carbons Production and Its Application for Supercapacitor Electrodes"

_polymers, 2023, doi:10.3390/polym15071760_

Round 1
Reviewer 1 Report
In this work, the authors synthesized the activated carbon obtained from the chemical activation of bacterial nanocel-17 lulose (BCN) using fique juice as a culture medium and investigated their performance for supercapacitor. I think it can be published after addressing the following questions.
1. Please check the scale bar in Figure 2 and 3.
2. What does the “gravimetric capacitance, Ceis, Ccv, Ccp” refer to? Please cite the relevant literatures and distinguish them.
3. There is lots of “Error! Reference source not found” in paper. Please check.
4. The arrangement of the graph in paper needs to be improved?
5. English needs to be polished. Please check whole paper.
6. Some relevant literatures for activated carbon should be referenced, such as Energy & Fuels, 2020, 34(12): 16885-16892, J. Power Sources, 2023, 554: 232347.
Author Response
We acknowledge the reviewers for their observations and recommendations; they were significant in improving our work. The questions are labeled in red, the answers are in black, and the changes in the manuscript are in blue.
Please see the attachment.

Reviewer 2 Report
In this work, the author reported "Bacterial nanocellulose from K. Medellinensis in fique juice for activated carbons production and its application for supercapacitor electrodes" and systematically characterized using various physiochemical techniques. Therefore, I recommend this work for publication in the polymers journal. However, some of the major concerns should be addressed before proceeding with further actions.
1. Authors should include FTIR and XRD of the prepared samples.
2. Authors should calculate Id/Ig value in the Raman spectrum.
3. In the title authors should give the full form of K. Medellinensis.
4. Authors should include the results of Three electrode measurements (e.g. CV, GCD)
5. Authors should include an equivalent circuit fit diagram in Nyquist plot.
6. Authors should cite the references properly, Many places it seems to be confused.
Author Response
We acknowledge the reviewers for their observations and recommendations; they were significant in improving our work. The questions are labeled in red, the answers are in black, and the changes in the manuscript are in blue.
"Please see the attachment."

Round 2
Reviewer 1 Report
The authors have addressed the points that I raised.
Author Response
Dear reviewer, Thank you!